# Development of Foot-and-Mouth Disease Vaccines in Recent Years

**DOI:** 10.3390/vaccines10111817

**Published:** 2022-10-28

**Authors:** Zhimin Lu, Shu Yu, Weijun Wang, Wenxian Chen, Xinyan Wang, Keke Wu, Xiaowen Li, Shuangqi Fan, Hongxing Ding, Lin Yi, Jingding Chen

**Affiliations:** 1College of Veterinary Medicine, South China Agricultural University, No. 483 Wushan Road, Tianhe District, Guangzhou 510642, China; 2Guangdong Laboratory for Lingnan Modern Agriculture, College of Veterinary Medicine, South China Agricultural University, Guangzhou 510642, China; 3Key Laboratory of Zoonosis Prevention and Control of Guangdong Province, Guangzhou 510642, China

**Keywords:** foot-and-mouth disease, inactivated vaccine, adenovirus vector vaccine, synthetic peptide vaccine, adjuvant

## Abstract

Foot-and-mouth disease (FMD) is a serious disease affecting the global graziery industry. Once an epidemic occurs, it can lead to economic and trade stagnation. In recent decades, FMD has been effectively controlled and even successfully eradicated in some countries or regions through mandatory vaccination with inactivated foot-and-mouth disease vaccines. Nevertheless, FMD still occurs in some parts of Africa and Asia. The transmission efficiency of foot-and-mouth disease is high. Both disease countries and disease-free countries should always be prepared to deal with outbreaks of FMD. The development of vaccines has played a key role in this regard. This paper summarizes the development of several promising vaccines including progress and design ideas. It also provides ways to develop a new generation of vaccines for FMDV and other major diseases.

## 1. Introduction

In countries with highly developed animal husbandry, foot-and-mouth disease (FMD) is still feared as a worldwide viral disease of animals, resulting in the death of young animals and the reduced productivity of adult animals, which lead to huge economic loss and social consequences. It is a highly contagious transboundary disease that mainly affects domesticated animals such as cattle, swine, sheep, goats and buffalo, as well as about 70 other cloven-hoofed wild animals [1,2]. Although cattle are the main host, swine are susceptible to some strains too. The disease is caused by infection with the foot-and-mouth disease virus (FMDV). FMDV mainly infects the host through untreated contaminated food products, such as swill. The virus first invades the oro-pharynx in swine [3] (nasopharynx in cattle) and then spreads systemically, forming vesicles from the mouth to the interphalangeal space to the breast, nipple and foot [4]. At the same time, infected animals show clinical characteristics such as high body temperature, excessive salivary secretion and reduced milk production. They are prone to secondary infections and lose weight, resulting in a long-term loss of productivity. The disease spreads easily from animal to animal, but mortality is low because the infection is usually cleared within two weeks. Most deaths occur in young animals with myocarditis, i.e., degeneration of the heart muscle [2,5]. In endemic areas, young animals may also acquire resistance through maternal antibodies to reduce mortality. Swine are thought to be one of the important factors in the spread of foot-and-mouth disease because one animal emits as much aerosol as 3000 cows in a short period of time and the virus travels a long distance in air [6,7]. In cattle, buffalo and sheep (but not swine), infection may persist beyond the acute stage due to low levels of infectious virus present in the oropharynx [8,9].

The foot-and-mouth disease virus is a member of the picornavirus family. The genome of FMDV, which is over 8000 bases in length, includes a large open reading frame (ORF) encoding a polyprotein that is processed into mature polypeptides. The structural proteins that form the icosahedral capsid of the virus—VP1, VP2, VP3 and VP4—are encoded by genes 1D, 1B, 1C and 1A, respectively. Non-structural proteins encoded by genes 2A, 2B, 2C and 3A, 3B, 3C^pro^, 3Dpol and L^pro^ [10], are mainly responsible for FMDV maturation and replication. Moreover, 5′and 3′ untranslated regions (UTRs) are also important for the replication and translation of viral genomes [11] (Figure 1). The viral proteins commonly used in vaccines are VP1, VP2, VP3, as well as 3A and 3D in the context of assembled capsids or as antigenic peptides.

Due to the high variability of the virus, there are seven different serotypes of foot-and-mouth disease virus: O, A, C, SAT (Southern African territory) 1, 2 and 3, and Asian-1. Different FMDV serotypes have a tendency to recur within a geographical area with emergence and transmission cycles that may affect multiple countries. Therefore, the World Organization for Animal Health (OIE) also divides the epidemic regions into seven sections (Figure 2). Except for Asian-1, each serotype is geographically limited and endemic within its own region [12]. Serotypes may spread to other regions, as seen with SAT-2, which has also become endemic in Egypt in recent years [13]. Serotypes O and A have a wide geographical distribution. However, the number of infections with serotype A has decreased significantly and infections with serotype O have been sporadic in recent years. Foot-and-mouth disease is currently endemic in several parts of Asia, large parts of Africa and the Middle East. Australia, New Zealand and Indonesia are called “foot-and-mouth disease-free countries” because the disease has disappeared due to the use of traditional vaccines that chemically inactivate the virus. Some countries do not vaccinate because of the high potential antigenic diversity of the virus, and in order to maintain foot-and-mouth disease-free countries and promote trade in animals and animal products. However, foot-and-mouth disease is a transboundary animal disease (TAD) which can occur occasionally in any typical free zone. Diversity of species and modes of transmission, high infectivity, rapid replication rates, high levels of viral excretion, increased international trade, rapidly changing environments and human activities could all lead to a resurgence of the FMDV [14]. Once an outbreak occurs, countries will need to reintroduce vaccines to control the spread of the disease. However, this is controversial and by no means risk-free. As we know, the earlier the vaccine is given before an outbreak, the better its protection is likely to be [15]. In order to control the disease, measures such as preventive vaccination or containment policies must be considered. This paper reviews the research progress of foot-and-mouth disease vaccines and provides new approaches for the development of FMD vaccines.

## 2. Inactivated Virus Vaccine

At present, the commercial FMD vaccine is an inactivated virus vaccine, which was also the earliest used. As early as the 1930s, people began to use formalin to kill living viruses. In the mid-1960s, the use of formalin-inactivated FMDV as a vaccine, cultured in hamster kidney (BHK) cells, dramatically reduced the prevalence of FMD in some European countries. By the 1970s, the number of outbreaks in Europe became very low. Following a ban on foot-and-mouth vaccination in the early 1990s by the European Union (EU), emergency vaccination in the event of an outbreak was approved [16]. All of this shows that inactivated vaccines are important for preventing foot-and-mouth disease. FMDV is highly sensitive to neutralizing antibodies produced in response to whole inactivated viruses and antibody titers are closely related to protection. Zhang et al. divide the antibody levels of challenged animals into three zones: the “white zone”, the “gray zone” and the “black zone”. Animals in the white zone have high levels of antibodies and are likely to be completely protected from FMDV infection. Animals in the black zone have very low levels of antibodies and are susceptible to infection. The vaccinated animals in the gray zone have intermediate antibody titers, making it difficult to predict the level of immune protection. The antibody levels of the gray zone are associated with the antigen content of the vaccine. Low antigen levels may stimulate the production of fewer antibodies.

In addition, virus particle integrity also plays a very important role in the immune response induced by the swine FMD vaccine. The complete virus particle of FMDV is 146S [17], and is composed of 12 pentamers (12S). Each pentamer is composed of five propolymers (5S) and each propolymer is composed of structural proteins (VP1, VP2, VP3, VP4). The efficacy of the inactivated FMDV vaccine mainly depends on the integrity of the virus particles. Empty capsids (75S) provide protective immunity in vaccinated guinea pigs, but less effectively than complete virions (146S) [18]. Pentamers show poor immune protection because they can induce the production of non-protective antibodies against internal epitopes. However, the most protective intact capsids are highly unstable, degrading to less immunogenic 12S subunits at moderate temperatures or in the presence of weak acids [19,20]. Therefore, stability is a key factor in the production of foot-and-mouth disease vaccines. Of the seven serotypes of FMDV, O and SAT are particularly unstable. Improved methods are needed to maintain the stability of virus particles.

One of the most important issues in the production of inactivated vaccines is the selection of pandemic strains because of the lack of cross-protection between different serotypes of FMDV. Even protection between certain strains within the same serotype is incomplete [18,21]. According to a study by Pirbright Institute, there have been no reports of disease caused by FMDV serotype C anywhere since 2004, and it can be said that this serotype is now extinct outside the laboratory [22,23]. The efficacy of the vaccine pool in the event of an emergency outbreak is predicted by in vitro vaccine matching tests, which are usually based on the r1 value. The r1 value (relationship coefficient) is the ratio of the serum virus neutralization (VN) titer against a heterologous strain to the serum VN titer against a homologous strain. In addition to the r1 value, the potency of the vaccine against different strains within the same serotype should also be considered [24]. Only by selecting the effective inactivated vaccine strain can clinical disease be better prevented. In susceptible swine, FMDV serotype O is more likely than other serotypes to mutate, thus posing the risk of an emergency outbreak. The O/ME-SA/Ind-2001e vaccine strain has high immunogenicity and extensive antigen coverage [25], especially in some Asian countries, and can be used to protect against the emerging FMDV. It is actually one of the more common and widespread strains in recent years, according to reports published by the OIE. Of course, we should also take into account other conditions that can contribute to increased coverage, such as vaccine formulations and vaccination schedules. Meanwhile, for FMDV serotype A, laboratory evidence has shown that Malaysia 97 and A_22_Iraq 64 emergency foot-and-mouth disease vaccines could provide good protection against serotype A ASIA/G-IX/SEA-97 lineages [26]. He et al. isolated NAb (R50) that neutralized serotype O and A FMDV from a recovered natural bovine host by using a single-cell antibody isolation technique. They identified the neutralizing antibody in a complex with the virus by cryo-electron microscopy at high resolution. They confirmed its binding to a highly conserved region in the O and A serotype [27].

The development of suspension cell cultures can facilitate the production of inactivated virus vaccines. Currently, the continuous cell line for the production of FMDV is BHK-21, which can be cultured to high density in large volumes, with a high virus yield. Serum-free medium is preferred for the culture of BHK-21 to eliminate pathogens from animal component sources and to show consistent basic cell characteristics. At the same time, scientists have struggled to figure out how to produce higher viral titers with the lowest possible cell density. For example, CDM-2 is added to the culture medium [28]; another approach is through cellular modification, by knocking out genes such as HDAC9 that inhibit virus replication [29] or by expressing integrin αvβ6, improving the degree of the “match” between the virus and the cell so that more strains can reproduce on cell lines and produce higher virus yields [30].

## 3. Virus-Like Particle Vaccine

Although inactivated virus vaccines are recommended by the OIE for the control and eradication of FMD, there are still some deficiencies in vaccine production that lead to sporadic and even severe outbreaks of FMD. Virus-like particles (VLPs) are large particles composed of one or several structural proteins of the virus which do not contain viral nucleic acid and cannot replicate, and have an overall structure similar to that of virus particles. As an ideal substitute for the traditional inactivated virus in vaccine production, VLPs not only retain the spatial conformation of natural virus particles with epitopes that stimulate the production of neutralizing antibodies, but also have certain safety. It has been proven experimentally that VLPs stimulate dendritic cells in the same way as inactivated FMDV (iFMDV) and induce humoral immunity at the same time [31].

There are generally two design approaches for VLP vaccines. Empty capsids can be produced within the vaccinated host, from capsid genes carried by a viral vector such as adenovirus. Alternatively, empty capsids can be produced in culture (bacteria, mammalian cells, insect cells or plants) and then given as a vaccine [32].

### 3.1. Adenovirus Vector Vaccine

An adenovirus vector vaccine for FMD is licensed as a candidate vaccine in the United States for emergency use in outbreaks of FMD due to FMDV type A [33,34], and is currently the most promising alternative to inactivated virus vaccines for worldwide use. The following objectives can be achieved by the production of FMD VLPs in vivo with human adenovirus type 5: Firstly, the vector particles can be rapidly internalized by host cells. Second, multiple specific immune responses can be activated. FMDV peptides will be presented in a complex with MHC-I molecules on the surface of host cells infected with the vector, and stimulate a CD8+ cytotoxic T-cell response. Third, infection in immunized animals can be avoided.

Adenovirus is a double-stranded DNA virus with a genome of about 34–43 Kb and more than 110 types (human adenoviruses, refer to: http://hadvwg.gmu.edu/ accessed on 10 October 2022). In 1999, the first adenovirus-based vaccine against FMD was produced. The vaccine uses an E1A/E1B deleted adenovirus, with sequences encoding the FMDV capsid P1-2A and the viral 3C protease that mediates the processing of viral proteins and the formation of neutralizing antibodies [35]. It has been shown to be effective in protecting swine from clinical disease following direct contact with an infected animal [36]. P12A3C boxes from different strains of FMDV have different effects on animals immunized with adenovirus vectors. On trial, Moraes et al. found that P12A3C boxes from FMDV A24/Cruzeiro/BRA/55 encoded in human adenovirus type 5 (Ad5-A24) provided early protection against homologous FMDV attacks in swine and cattle [37]. The Ad5-A24 model was thus established, and then it was fully developed into a commercial product. Pena et al. modified the Ad5-A24 vector by including FMDV sequences encoding non-structural proteins [38]. It should be noted that if additional sequences encoding additional non-structural proteins are included in the recombinant adenovirus vector genome, the advantage of differentiating infected from immunized animals (DIVA) may be lost. For the most common FMDV serotype O, a human adenovirus 5 (Ad5-O1Man) with FMDV O/Manisa/TUR/69 as an antigen has also been explored. This FMDV antigen is expressed even more efficiently than the A24 antigen in swine [39]. However, the adenovirus vector vaccine for FMDV serotype O is inferior to the vaccine for serotype A because fewer FMDV VLPs are produced in the vaccinated animal [40]. The adenovirus vector vaccine for O1/ Campos/Brazil/58 is even less effective than other subtype O vaccines [41]. In a recent study, Micaela et al. used the optimized Ad5 vector-Ad5[P_VP2_]_OP_ to effectively improve the immunogenicity of the FMDV O1/Campos vaccine [42]. The optimized model harbors the foreign transcription unit in a leftward orientation relative to the Ad5 genome and drives the expression of the FMDV sequence from the optimized cytomegalovirus (CMV) enhancer–promoter. The amino acid substitution of S93F/Y98F in the VP2 protein coding sequence stabilizes the VLPs. The vaccine has only been tested in mice so far.

### 3.2. Phage Vaccine

T7 and T4 phages have been used in VLP vaccine development. More of this work has been undertaken with T7 phage. T7 is a lytic phage in the family of Brachyviridae of the order Urophage. Its head is composed of linear dsDNA (39,936 bp) and six capsid proteins, including gp10A, gp10B, GP8, GP11, GP12 and GP17, with a total of 415 proteins in each head [43]. Peptides of FMDV VP1, when displayed on the phage capsid, induced a specific antibody response in mice [44,45], suggesting that such particles may have potential as a vaccine. The stability of T7 phage would be a distinct advantage in reducing the storage and shipping costs of such vaccines.

### 3.3. Nucleic Acid Vaccine

Nucleic acid vaccines are a promising and attractive option because they are safe, stable and easy to administer, produce and store. The pcDNA3.1/P12A3C expression plasmid-encoding proteins of FMDV can be packaged with capsid proteins in vitro to form VLPs that can be used as vector particles to deliver the expression plasmid to target cells in the host. The expression of FMDV proteins then stimulates immunity through the MHC I and II pathways to produce antibodies as well as CD4+ and CD8+ T cells. These DNA-loaded VLPs function as DNA vaccine carriers that protect the plasmid DNA from degradation by host enzymes. Moreover, they enhanced the level of a specific antibody and prolonged its duration compared to VLPs without DNA and to plasmid DNA alone in a guinea pig model [46]. Plasmid DNA vaccines given alone are not highly effective, requiring three doses to achieve effective antibody levels in pigs [47,48], though co-injection with plasmid-encoding GM-CSF induced a stronger response in cattle [46] and swine [47].

### 3.4. E. coli Expression System

*Escherichia coli* (*E. coli*) is the most common expression system for VLPs of FMDV. The FMDV serotype O capsid protein containing a small ubiquitin-like modifier (SUMO) is expressed in *E. coli* by optimal tandem arrangement (VP0-VP3-VP1) [49]. In this way, assembled FMDV VLPs can expose multiple epitopes and have a similar size to the original FMDV. While effectively inducing the humoral and cellular immune responses specific to FMDV in swine, the efficiency was also increased with increasing dose. In order to increase the stability of the VLPs, Li et al. modified selected amino acids then screened the modified VLPs for increased hydrophobic force inside the capsid, better yield of VLPs, and immunogenicity of the VLP vaccine [50]. In addition, some live attenuated bacterial expression systems, such as Salmonella typhimurium [51] and Lactococcus lactis [52], are used as oral vaccines for the expression of FMDV proteins in the vaccinated host.

### 3.5. Mammalian Expression System

In recent years, the expression of VLPs in mammalian cells has also become popular [53]. Transient gene expression (TGE) is used generally because of its characteristics of multiple copies, repeatability, low cost and rapid production of a large number of proteins in a short time [54]. For TGE, P12A sequences of different FMDV can be easily cloned into pTT5 vectors. Unmodified polypeptide sequences enable the platform to quickly adapt the new topologies and subtypes, resulting in an immune response to different FMDV serotypes. For example, VLPs produced by the TGE of A2001 Argentina strain and O1Man can trigger an immune response in immunized animals, including swine and cattle [53,54]. TGE may also address problems related to virus adaptation in cell culture, but further studies are needed to demonstrate the recombinant expression of more serotypes.

### 3.6. Chimeric Vaccine

The development of the chimeric VLP vaccine is also an approach that induces a strong specific humoral response. Epitopes of FMDV displayed on VLPs composed of hepatitis B core protein or HIV Gag polypeptide or rabbit hemorrhagic disease virus (RHDV) VP60 induce the formation of FMDV-specific antibodies [55,56,57,58]. Similar VLPs can also be taken up by dendritic cells to expose T-cell epitopes through the histocompatibility complex (MHC) I pathway, thus initiating a cytotoxic T-cell response [59,60,61]. It may be expected that VLPs with T-cell epitopes of FMDV could induce a cytotoxic T-cell response, as seen for other T-cell epitopes expressed as part of RHDV VLPs [59,60].

## 4. Synthetic Peptide Vaccine

The VP1 G-H ring is the most important protective antigen. FMDV is recognized by its RGD motif (amino acid residue 130–160), which is the host cell integrin receptor binding site and plays an important role in the production of neutralizing antibodies to FMDV. Thus, VP1 has been used in one of the first attempts to produce FMD peptide subunit vaccines [62,63]. T-cell epitopes were identified in VP4 (internal, highly conserved, amino acid residue 20–34) [64,65] and in other structural proteins in cattle and swine. These epitopes are easy to express in peptide vaccines and the epitope sequence can be changed to represent the appropriate strain in an epidemic scenario.

The first peptide vaccines represented single linear epitopes. More recently, the multiple-antigen peptide system (MAP) (Figure 3) has been developed. MAPs are dendritic polymers with lysine dendrimers exposed as core and dendritic arms with multiple antigenic epitopes. This dendritic peptide copolymer is more immunogenic than a simple linear juxtaposition of B and T epitopes and is moderately resistant to enzyme digestion. In contrast, VP1 is degraded easily by trypsin.

In the past, B4T was designed in view of the fact that the seven FMDV serotypes reflect antigenic diversity and there is no cross-protection among the serotypes. B4T is composed of an immunodominant T-cell epitope (3A) that covalently connects four different VP1 GH-loop sequences in the form of dendritic branches so that more B-cell epitopes can be connected and the vaccine strain can better match the circulating virus [66]. However, it was soon found that a simple structure of B2T constructs with only two branches of B-cell epitopes was better at inducing specific responses [67]. At the same dose and time interval, the release of the antibody and IFN-γ release were increased to give stronger protection when the swine were infected [68]. Along with adenovirus vector vaccines, this peptide vaccine is one of the few to show that a single dose can replace conventional FMDV vaccines for protection against disease. Single-dose vaccines not only help reduce costs, including vaccine materials, labor and logistics, but also facilitate rapid vaccination in emergency situations which require a fast, low-cost and flexible response to the virus.

T-cell and B-cell epitopes are the key factors in the development of synthetic peptide vaccines. The non-structural 3D proteins (amino acids 56–70) in FMDV have been used in synthetic peptide vaccines [69,70]. Swine which were inoculated with B2T-3D induced the same level of neutralizing antibodies and IFN-γ as B2T-3A, and inoculation with B2T-3D induced even more IFN-gamma [71]. The cross-neutralizing antibody titer induced by B2T-3D was higher than that included by B2T-3A. The B2T-TB2 peptide, with six epitopes, is formed by two B2T molecules and can induce a high level of protective mucosal IgA response in domestic swine infected with FMDV [72]. The structure also induced strong humoral immunity against classical swine fever virus (CSFV), a pestivirus within the family Flaviviridae

Recent findings show that specific MHC alleles and haplotypes affect the efficacy of peptide vaccines containing a limited number of T-cell epitopes [73]. Porcine MHC (SLAs, swine leukocyte antigens) is one of the most well studied major histocompatibility complexes outside of humans [65]. Moreover, 266 SLA-I (SLA-1, -2, -3), 227 SLA-II (DR, DQ, DM, DO), 2 SLA-related alleles and 2 non-SLA-related alleles have been found in swine (Table 1). These alleles have been subdivided into different haplotypes by high-resolution DNA sequencing. As of July 2019, there were 73 independent class I haplotypes and 51 class II haplotypes [74]. Low-resolution (Lr) haplotype analysis showed that SLA-II contributed to the number of specific SLA-II-restricted T cells induced by B2T dendritic molecules in swine, but more data are needed to support this [73].

## 5. Adjuvant and Delivery System

Successful vaccines depend on safe and effective adjuvants, as well as appropriate delivery systems and route of administration. Adjuvants used to date include mineral oil (Montanide Isa-206, ISA-201), aluminum hydroxide, saponins (Quil-A), Toll-like receptor ligands (targeted pattern recognition receptors), cytokines (IFN-A, IFN-G, IL-1, IL-2, IL-15, IL-18 and GM-CSF) and liposomes. There are also new and effective adjuvants on the horizon (Table 2).

Oil adjuvants are more effective than other adjuvants in swine, causing stronger immune responses. At the same time, they can also cause serious side effects such as hemolysis, swelling or necrosis at the injection site. Oil-adjuvant vaccines induced strong immunity and were less interfered with by colostrum antibodies, but the widely used Montanide ISA-206 has been found to accelerate the degradation of immunologically active 146S particles at the oil–water interface [76].

Aluminum hydroxide induces the typical antibody-mediated Th2 response rather than cell-mediated Th1 immunity [77]. It also stimulates high levels of lgE production, which can lead to hypersensitivity and neurotoxicity with a strong inflammatory response at the injection site. Saponin was combined with aluminum hydroxide (alum) to effectively compensate for the lack of cellular immunity [78,79]. Saponins also provide an advantage when mixed with oil adjuvant. In cattle, a saponin-oil emulsion increased the level of neutralizing antibody with a lower dose of antigen (146S particles) [80]. Reducing the required dose of viral antigen represents a significant cost saving in vaccine production.

Humoral immunity to FMDV develops rapidly but is short-lived [81], even in the case of iFMDV emulsified with adjuvant [82]. Although the immunopotentiator CVC1302 can enhance humoral immunity [83], it is increasingly recognized that cellular immunity mediated by CD8+ and CD4+ T lymphocytes is equally important for adaptive protection in immunized animals. Since an enhanced cellular immune response can be more beneficial in some cases, the direction of adjuvant development has been shifted to stimulate the Th1 and Th2 responses to antigens as much as possible [84,85]. Antigens need to be taken up, processed and presented by dendritic cells (DCs) in association with MHC molecules. Some agonists have been shown to be recognized by CD4+ and CD8+ T cells and induce DC maturation in the process. Mature DCs are recognized by CD4+ and CD8+ T cells, resulting in both humoral and cellular immunity. TLR4 agonists are considered to be the main ligand to activate DCs. In recent years, there have been many examples of TLR4 agonists being used as adjuvants. Coating virus-like particles with dimethyldioctadecylammonium bromide (DDA)-based cationic liposomes and monophosphate liposome A (MPL) promote Th1 responses and even multifunctional T-cell immune responses such as Th17 [86]. Lei et al. constructed the recombinant fusion protein HAO-HBHA. They connected the multi-epitope immunogen HAO of FMDV serotypes A and O in tandem with heparin-binding hemagglutinin (HBHA) of *Mycobacterium tuberculosis*, a novel TLR4 agonist. This modification not only increased the expression of IL-4, IL-6, IL-10 and IL12p70, but also increased the stability and solubility of HAO protein [87].

Chinese herbal medicine is a material that cannot be ignored in adjuvants. Xu et al. used panax ginseng stem and leaf saponins as immune enhancers. Carbohydrate groups on the saponin molecule can interact with receptors on antigen-presenting cells (APCs) to promote a Th1 response [88]. The crude polysaccharides of *Cistanche deserticola* (CPCD) can activate DCs through TLR-2 and TLR-4, inducing the activation of MAPK and NF-κB pathways [89]. CPCD can trigger FMDV specific immune responses such as increased specific antibodies, splenocyte proliferation, T-cell subsets, IFN-γ, CTL and DC activation at the optimal dose of 400 μg in mice. The synergistic effect of *Artemisia rupestris* L. (AEAR) and ISA-206 oil adjuvant can increase serum antibody titer, enhance cytokine secretion and stimulate long-term immunity [90]. In addition, the preparation of VLP vaccine using *Achyranthes bidentata* polysaccharide (ABP) as an immunostimulant can promote the proliferation of splenic lymphocytes and increase specific antibodies to enhance the immune effect, and the preparation can be stored at room temperature for a long time [91].

The non-coding RNA (ncRNA) of FMDV also has potential as a vaccine adjuvant. ncRNAs are short RNA sequences whose structures are similar to those within the non-coding regions of the FMDV genome. The 3′ NCR, as well as the IRES and S domains from the 5′ NCR, induced stronger and longer-lasting B- and T-cell responses, enhancing protection via the inactivated FMD vaccine in swine [92]. However, it is important to note that IRES transcripts may reduce the specific humoral response to MAP (multiple antigen peptide) vaccines [93].

In some cases, delivery systems are not easily distinguished from adjuvants because they can also stabilize and enhance antigens in addition to delivering them to the appropriate target cell. Nanoparticle polymers (NPs) are the most promising delivery systems currently in use because of their unique physicochemical properties such as controllable shape and size, with the advantages of large specific surface area, multiple-surface active centers and high reactivity. Mesoporous silica nanoparticles (MSNs) [94], chitosan nanoparticles (CS) [95], gold nanoparticles and L-Lactide-co-glycolic acid (PLGA) [96] have received significant research attention in the biomedical field and have been shown to enhance immune response.

In recent years, mesoporous silica nanoparticles have become the most popular transport system. Dendritic mesoporous silica nanoparticles (DMSNs) have a unique central–radial pore structure and a greater loading capacity. An et al. used DMSNs with B2T synthetic peptide and showed prolonged release when the complex was coated with serum proteins [97]. Monodisperse silica microspheres have good uniformity and controllable size, but they are difficult to degrade under natural conditions. Yu et al. used disulfide (S-S) groups to destroy the strong skeleton (-Si-O-Si-) in the original microspheres and avoid the accumulation of mesoporous silica nanoparticles in the reticuloendothelial system [98]. On this basis, Yin et al. loaded imiquimide (IMQ) into the degradable mesoporous silica micro-spheres in order to control the release rate of FMDV and reduce the toxicity of IMQ to immunized animals [99].

As a new reagent, chitosan (CP) has been used in the development of many adjuvants because it can be functionalized by different functional groups. There are many His residues on the capsid of FMDV. Divalent metal ions can neutralize His residues, so that iFMDV can be well adsorbed on the zinc-chelated chitosan particles (CP-PEI-Zn) in order to increase the thermal stability of the inactivated virus and promote the humoral and cellular immune responses of mice [100]. Bionic chitosan hydrogel nanoparticles were used to simulate the flexibility of the pathogen’s configuration and deformation to increase the contact area between the vaccine and cells for the better delivery of antigens to APCs [101]. Coating PLGA nanoparticles with CP (PLGA-NPs) could improve the immune response to mucosal inoculation [102]. Zheng et al. used chitosan-coated PLGA-NPs mixed with amino-functionalized mesoporous silica nanoparticles, loaded with FMDV recombinant plasmid, for intranasal vaccination along with CpG oligodeoxynucleotides encapsulated in chitosan-coated PLGA-NPs as adjuvant [103]. Systemic and mucosal immune responses, including specific IgA, were induced in guinea pigs. Challenge experiments showed protection against the systemic spread of the virus in vivo following the injection of the virus into one leg [103].

Gold nanocages (AuNCs) had little biological toxicity in vitro and in vivo, and increased the uptake of VLP via the BHK-21 and RAW264.7 cell lines [104]. The combination of VLP and AuNCs significantly promoted the proliferation of CD8+ T cells and the release of immune-related cytokines [104].

Layered double hydroxides (LDHs) are structures with alternating layers of hydroxide molecules separated by interspace layers filled with anions and water. These anions can be exchanged for other molecules, including biological molecules, meaning that LDHs can be loaded with biological molecules and used as delivery vehicles [105]. The properties of LDHs, including the sustained delivery of their load, make them attractive as potential vaccine carriers [106]. When used to immunize mice by subcutaneous injection, LDHs loaded with inactivated FMDV particles induced high and sustained antibody levels, following an initial delay [107]. In pigs, the LDHs loaded with inactivated FMDV particles induced specific neutralizing antibodies at levels comparable to levels achieved with Montanide ISA-206 adjuvant [107].

In addition, some immune-stimulating particle adjuvants (ISPA) [108], ferritin nanoparticles [109], solid lipid nanoparticles (SLN) [110] and bio-mineralized nanomaterials [111] have also been proven to enhance the stability of the antigen, as well as the proliferation and the differentiation of central memory T cells and effector memory T cells.

**Table 2 vaccines-10-01817-t002:** A review of adjuvant and delivery systems against foot-and-mouth disease virus (FMDV) published in the last few years.

Type	Adjuvant or Delivery System	Mechanism	Applicable Vaccines
	Saponin	The imine carbonyl group formed contributes to T-cell activation (inducing Th1/Th2 response) and permeabilizes cell membranes [78,79,80]	Adenovirus vector vaccine
	CAvant ^®^ SOE (CA V AC, Daejeon, Korea)	Delivery of antigens to APCs or by direct stimulation of immune cells [112]	Inactivated viruses
Agonists	Cationic liposomes and monophosphate liposome A	VLP is encapsulated in a cationic liposome and/or MPL based on DDA [86]	VLP vaccine
Agonists	Heparin-binding hemagglutinin (HBHA)	The multi-epitope immunogen HAO of serotype O and A FMDV was combined with HBHA, a novel TLR4 agonist [87]	VLP vaccine
Agonists	CVC1302	Contains three PRR agonists that can increase B-cell numbers to increase antibody response [83,113]	Multi-epitope recombinant vaccine
Chinese herbal medicine	Panax ginseng stem and leaf saponins	The carbohydrate groups on the saponin molecule can interact with receptors on the APCs, and the acyl domain can facilitate the entry of antigens into the APCs [88]	Inactivated viruses
Chinese herbal medicine	Crude polysaccharides of *Cistanche deserticola* (CPCD)	DCs were activated by TLR-2 and TLR-4, and MAPKs and NF-κB pathway were induced [89]	Inactivated viruses
Chinese herbal medicine	*Artemisia rupestris* L., (AEAR)	Increase serum antibody titers, enhance cytokine secretion, and stimulate T-cell-mediated immune responses [90]	Inactivated viruses
Chinese herbal medicine	*Achyranthes bidentata* Polysaccharide (ABP)	The stable polysaccharide nanoemulsion delivery system can better deliver antigen and promote immune enhancement [91]	VLP vaccine
	Noncoding synthetic RNAs	IRES, S and 3′NCR domains transcribed in vitro from plasmids induce a powerful antiviral response [92]	Inactivated viruses
Nanoparticle polymers	Mesoporous silica	Unique center–radial hole structure for greater load capacity and control of FMDV release rate [97,98,99,103]	Inactivated viruses
Nanoparticle polymers	Chitosan (CP)	The flexible configuration and deformation of the vaccine particles can increase the contact area with cells [100,101,102]	VLP vaccine and inactivated vaccine
Nanoparticle polymers	Gold nanocages (AuNCs)	Proteins can bind to gold nanomaterials by electrostatic interaction, hydrophobicity and Au-S bond cooperation [104]	VLP vaccine
Nanoparticle polymers	Layered dihydroxide (LDH)	These particles, with interspace layers that can be loaded with antigens, provide improved and sustained delivery of antigen in vivo [107]	Inactivated viruses

## 6. Conclusions

Foot-and-mouth disease is the first disease for which the OIE established an official list of disease-free countries which can be officially recognized as free of the disease either in their entirety or in defined zones and compartments. Some disease-free countries have stopped vaccinating. However, countries that have not received the vaccine may be under constant threat of foot-and-mouth disease invasion. If FMD catches on, it could become a major constraint on international livestock trade. Therefore, it is extremely important for us to prevent and control foot-and-mouth disease. As a picornavirus virus, the FMDV is consistent with other RNA viruses which have a high rate of genetic variation due to mutation and recombination during the process of replication. VP1 is most frequently affected by mutation, whereas VP4 is more conserved. A major challenge with the FMD vaccine appears to be that the available vaccine targeting one serotype of FMDV does not generate immune memory cross-reacting against other serotypes of FMDV. Given that infected swine exhale large amounts of virus that can travel up to 70–330 miles in the air [114], it is important that vaccines be effective in swine. Whereas the established vaccines, in use since 1930s, are inactivated virus particles, the newer vaccines in use represent subviral components in the form of adenovirus vectored vaccines, VLPs and synthetic peptides. In all cases, the immunogen is intended to elicit a protective response against multiple strains of FMDV. The duration of immunity is also a challenge. Whereas existing inactivated virus vaccines require repeated dosing, it is desirable that vaccines would provide protection after a single vaccination.

With newer vaccines that protect animals against multiple virus strains, we will be better prepared against the next outbreak of FMD for a longer period of time. Continued research should lead to further improvements in protection against FMD.

## Figures and Tables

**Figure 1 vaccines-10-01817-f001:**
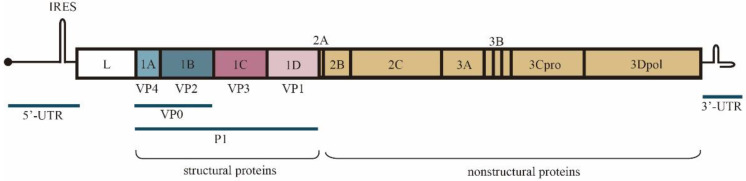
Production and processing of FMDV proteins.

**Figure 2 vaccines-10-01817-f002:**
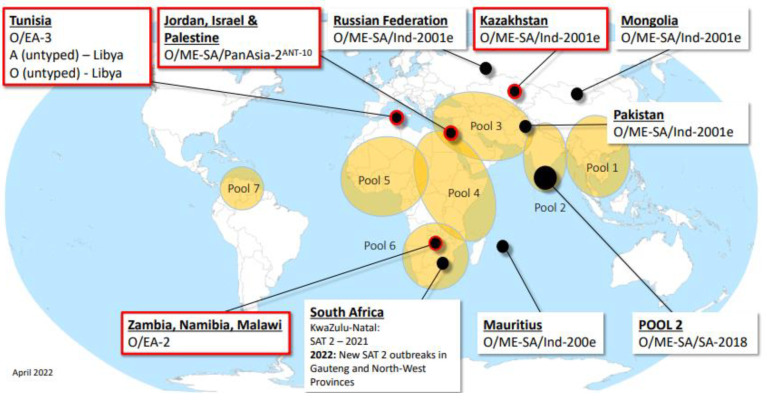
Recent FMD global outbreaks (new headline events reported January to March 2022 are highlighted) with endemic pools highlighted in orange. Source: WRLFMD. Map conforms to the United Nations World.

**Figure 3 vaccines-10-01817-f003:**
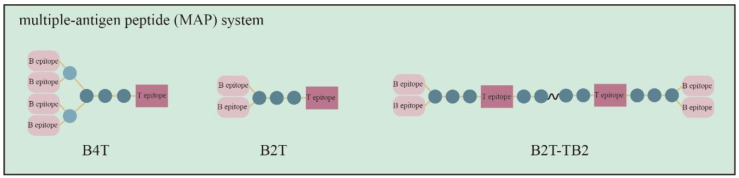
A multiple-antigen peptide (MAP) system model.

**Table 1 vaccines-10-01817-t001:** Numbers of confirmed swine leukocyte antigen (SLA) alleles and proteins [74,75].

Category	Locus	Allele	Protein
SLA class I (classical)	SLA-1	90	88
	SLA-2	97	94
	SLA-3	41	39
SLA class I (nonclassical)	SLA-6	10	10
	SLA-7	3	3
	SLA-8	5	5
SLA class I (unclassified)	SLA-12	6	6
SLA class I (pseudogene)	SLA-4	3	0
	SLA-5	4	0
	SLA-9	5	0
	SLA-11	2	0
**Total class I alleles**		**266**	**245**
SLA class II	DRA	14	6
	DRB1	99	92
	DQA	26	24
	DQB1	53	48
	DMA	7	5
	DMB	1	1
	DOA	2	2
	DOB1	3	3
SLA class II (pseudogene)	DRB2	12	0
	DRB3	5	0
	DRB4	1	0
	DRB5	1	0
	DQB2	1	0
	DQB2	1	0
	DYB	1	0
**Total class II alleles**		**227**	**181**
Other non-SLA genes	MIC-1	1	0
	MIC-2	1	1
	TAP1	1	1
	TAP2	1	1
**Total SLA-related alleles**		**4**	**3**

## Data Availability

No new data were created or analyzed in this study. Data sharing is not applicable to this article.

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
