# Peer review of "Development of Foot-and-Mouth Disease Vaccines in Recent Years"

_vaccines, 2022, doi:10.3390/vaccines10111817_

Round 1

Reviewer 1 Report

This review provides an up-to-date summary of the current state of FMD vaccines, incorporating many references published within the last year or two.  The paper is dense and difficult to read because wording and sentence structure throughout the document are awkward and sometimes grammatically incorrect. Unfortunately, these issues often compromise understanding, forcing the reader to look up the corresponding reference in order to understand the point that the authors are trying to make. The paper has the potential to be an informative review once it is re-written to address these issues.

In an effort to be helpful, I have pointed out the problematic wording and made suggestions for improvement. Specific comments are listed here: 

Throughout the paper, bacterial names should be italicized.

Line 15 – vaccination with (not “of”)

Line 19 – the word “progress” is used twice.  I suggest replacing the first “progress” with “development”

              - the ways (delete “the”)

Line 20 – diseases rather than epidemics

Line 25 – open bracket “{“ at the end of the line looks strange.  Perhaps it’s a typesetting issue.

Line 31 – should be “some strains are pathogenic in swine”  (ie swine are sensitive to the virus not the other way around)

Line 34 – “systemically” not “systematically”

Line 35 – Is the word “paw” appropriate for swine and cattle?  It seems strange, usually applied to animals like cats and dogs.  Would “foot” be better?

Line 36 – “showed” rather than “were accompanied by”

Lines 42-45 – Suggested re-arrangement of words” “Swine are thought to be one of the important factors in the spread of foot-and-mouth disease because one animal emits as much aerosol as 3000 cows in a short period of time and the virus travels a long distance in air. “ 

      - A reference is needed to support this statement.

Line 47 – picornavirus family ie delete “es”

Line 48 – “encoding” rather than “including”

Lines 50-51 – problem with sentence structure – add a period after “respectively” and begin a new sentence “Non-structural proteins …… replication.”

Line 53 – It isn’t clear what is meant by “them”.  Delete “among them” (not necessary).

Line 55 – awkward and meaning not clear.  Do you mean “in the context of assembled capsids or as antigenic peptides”?

Fig. 1 – should IRSE be IRES ie internal ribosome entry site?

Line 60 – “geographical” should be one word, not hyphenated

              - awkward wording. Replace with “ …… a geographical area with emergence and transmission cycles that may affect ….”

Line 62 – “epidemic regions” than “epidemic area”

Line 63 – “plate” is a strange word here.  Do you mean “region”?

              - awkward sentence structure and sentences shouldn’t begin with “but”.  Instead “Serotypes may spread to other regions, as seen with SAT-2 ….. recent years”  

Line 64 – delete “Among these serotypes”.  Begin sentence with “Serotypes O and A …..”

Line 65 – prevalence (meaning incidence of disease due to those serotypes) being extensive doesn’t make sense.  Do you mean that their prevalence is higher than for other serotypes?

Line 66 – awkward wording.  Better to say “The number of infections with serotype (no “s”) A has decreased significantly and infections with serotype O have been sporadic in recent years.” 

Lines 70 - 71 – Do you mean “high potential of antigenic diversity”?  As for the rest of the sentence, it isn’t clear how the lack of vaccination maintains foot-and-mouth disease-free countries.   Is it related to testing for antibody which wouldn’t distinguish between infected and vaccinated animals?

Lines 80-82 – delete “tries to”.  Instead “This paper reviews the current state of vaccines against foot-and-mouth disease, including recent approaches to development of new and improved vaccines.”   This paper is focused on FMD vaccines; since other vaccines aren’t discussed, it isn’t appropriate to mention other epidemic diseases here.  

Line 87 – insert “virus” ie inactivated virus vaccines

Line 90 – not clear what is meant by “it shortened the time of the vaccine”.  Does “it” refer to the use of formalin?  What is meant by “the time of the vaccine”?

Lines 91-92 – delete “made”

                     -  The sentence needs to be re-worded.  The statement that virus was cultured in BHK (baby hamster kidney) cells would fit better at the beginning of the sentence – as is, it doesn’t make sense.  Suggested improvement: “ In the mid-1960s, the use of formalin-inactivated FMDV cultured in BHK cells, as a vaccine, dramatically reduced the prevalence of FMD in (not “of”) some European countries.”

Lines 93-95 – “Following a ban on ……….. an outbreak” isn’t a sentence.  Do you mean “Following a ban ……….. (EU), emergency vaccination was allowed in the event of an outbreak”?        

Line 99 – “have” rather than “has”

Line 100 – “three zones” rather than “three regional scope” to keep terminology consistent

               - use “FMDV” for “foot-and-mouth disease virus” for consistency

Line 101 – “antibody concentration of between 1 to 1.6 antibody concentration” doesn’t make sense. Are there units for antibody concentration (if so, state them) or is it relative (if so, to what?)

Lines 102 – 103 – “which ….. identified” doesn’t make sense. Do you mean that these antibody levels can’t be interpreted to mean that animals are protected?  or that it isn’t known what antibody levels correlate with protection?

Lines 101 and 103 – “zone” rather than “area” to be consistent with “zones” in line 99

Line 107 – “immune response” better than “immune serum”

Line 108 – “and is” rather than “which is”

                - propolymers (plural)

Line 109 – delete “a group of” – it’s confusing. A figure would be very helpful for the reader.

Line 111 – What is meant by “it was followed”? The rest of the sentence seems to say that empty capsids (75S) provided the most effective (ie greatest) protective immunity. Is that what the authors mean?

Line 112 – pentamers (plural)

Line 113 – “they” not “it”. 

  - Better to say “production of non-protective antibodies against internal epitopes”

Line 114 – “which is not conducive to the full development of T cells”.  It is not clear what is meant.

Lines 114 – 115 – delete “the truth is that”

                            - substitute “protective” for “powerful”

                           -  do you mean “capsids” as in line 111 or “complete virions”?   If “virions”, the statement contradicts line 111.

Lines 115 – 116 – “Degrading …… weak acids” is not a sentence.  Is it meant to be part of the previous sentence ie “The most protective intact capsids are highly unstable, degrading to less immunogenic 12S subunits (rather than sub-particles) at moderate temperatures or in the presence of weak acids?”

Line 117 – “stability …… is key technology” doesn’t make sense.  Do you mean that stability is a key factor in production of ……?

Line 119 – what is meant by “They should be selected through some treatment ….”?

               - Is “adjuvant treatment” an appropriate term?   Adjuvant is added to stimulate an immune response but does it stabilize the antigen?

              -  Do you mean that “improved methods are needed to maintain stability of the virus particles”?

Line 123 – end sentence after FMDV.  Start new sentence as “Even protection between certain strains within the same serotype is incomplete.”

Line 126 – “vaccine efficacy” – delete “vaccine” to avoid repetition

Lines 128 – 129 – “homologous” and “heterologous” strains are confusing here.  Do you mean “The potency of the vaccine strain against different strains within the same serotype should be considered”?

Line 130 – delete “correct and”

               - “……. effective inactivated vaccine of the strain” - delete “of the”

               - disease (singular) 

Line 131 – awkward.  Suggested: “FMDV serotype O is more likely than other serotypes to infect susceptible swine and mutate …… outbreak.”

Line 134 – use FMDV, rather than writing out the words, for consistency

Line 135 – change “which is known in” to “according to”

Line 140 – It is not clear what is meant by “heterogeneic attacks”.  The sentence does make sense if “against heterogeneic attacks” is deleted. 

Line 141 – finding conserved sequences among different serotypes is a potential key……

Conserved sequences don’t necessarily stimulate cross-protection. Note the many years of research into a universal influenza vaccine and we’re not there yet.

Line 143 – “from a recovered natural bovine host …….. by using a single cell ……… technique”.  Add the two “a”s.

Line 145 – “atomic resolution locus” seems strange. Do you mean “…….. of the virus in a locus of complex structure, as determined by electron microscopy at high resolution”?

Lines 145 – 146- not a sentence.  Do you mean “They confirmed its binding to a highly conserved region in the O and A serotype”?

Line 147 – Awkward and confusing wording. Suggested: “………. suspension cell cultures can facilitate the production of inactivated (not inactive) virus vaccines.”    Underlined words to be added.

Line 148 – awkward, as is.  Better to say “….. cell line for production of FMDV is BHK-21 ……”  The name of the cell line should be given as baby hamster kidney the first time the term BHK-21 is used (lines 91-92), not here.

 Lines 149-150 – What is meant by “high volume yield”?  Do you mean that cells can be cultured to high density in large volumes, with high virus yield?

 Lines 153-154 – “lowest” rather than “smallest”

                         - “Some material is added ………. to modify it”.    This statement is weak – be more specific or delete it. Alternatively, “Approaches to achieve higher yields include addition of agents such as (give examples) to the culture medium and modification of the cells by knocking out genes that inhibit virus replication or by expressing integrins to mediate interactions of more virus strains with the host cells.”

 Line 158 – Virus-like (upper case V)

 Line 159 – inactivated virus vaccines -   The word “virus” is important here; it is the virus, not the vaccine, that is inactivated.

 Lines 160-161 – FMD, as used earlier in the paper

                         - vaccine (no hyphen)

                         - “deficiencies” rather than “defects”

 Line 162 – inactivated virus (not vaccine) in vaccine production

 Line 163 – awkward wording.  Better “………. conformation of natural virus proteins with epitopes that stimulate production of neutralizing antibodies”

 Line 164 – “proven” not “proved”

 Line 167 – VLP not VLPs

 Lines 167 – 172 – the terms “endogenous” and “exogenous” production may not be obvious to many readers.  It would be better to use more direct language eg “……. vaccines. Empty capsids can be produced within the vaccinated host, from capsid genes carried by a viral vector such as adenovirus. Alternatively, empty capsids can be produced in culture (bacteria, mammalian cells, insect cells or plants) and then given as a vaccine.”

 Line 174 –Modify as follows:  “An adenovirus vector vaccine for FMD is licenced as …..”

               - How is the vaccine administered?   By intramuscular injection?

 Line 176 – “production” rather than “delivery”.   VLPs are made within the host following vaccination with the adenovirus particles.

 Lines 177 – 180 - “it” is confusing here.  Do you mean “the vector particles can be rapidly internalized by host cells” or “the VLPs made in cells infected with the vector particles can be rapidly internalized by other  host cells”?   Clarify.

                           - “immunized” cells doesn’t make sense.  Do you mean cells of the immune system?

                           - “to obtain the uptake and expression of required genes” is confusing and doesn’t contribute anything – delete.

                           - what is meant by “immune modes”?

                           - what is meant by “they” in “They can be presented …”

                           - presented by MHC class I molecules not to MHC class I molecules

Line 180 – it is not clear what is meant by “These genes” in this sentence.  Do you mean that FMDV genes are expressed in host cells infected with the vector?  If so, say so.

               - Do you mean that FMDV peptides will be presented in a complex with MHC-I molecules, on the surface of host cells infected with the vector, for recognition by CD8+ cytotoxic cells?  Clarify.

Line 181 - “innate response” here is confusing. Delete.  Since the vaccine is meant to induce humoral immunity (antibodies) and cell-mediated immunity, focus on those. 

Line 182 – what is meant by “it can also infect and express transgenic favourable factors in mucosal tissues”?  Does “it” mean “the vector particles”?  What do you mean by “transgenic favourable factors”?

How are these vector particles administered?  Will they reach mucosal surfaces (not tissues)?

 Lines 185 – 186 – This information should be supported by a recent reference.

                             - The number of human adenovirus types now exceeds 110.  See website for Human Adenovirus Working Group <hadvwg.gmu.edu>   Since designations beyond type 51 are based on DNA sequence rather than serological data (with neutralizing antibodies), they are types not serotypes.  When giving numbers of adenovirus types, it is important to specify human adenoviruses.  There are several hundred types, if adenoviruses affecting other species (eg animals, birds, reptiles) are included. 

                             - clarify “first adenovirus-based vaccine against FMD”  (Ad5-FMD is the vaccine strain, not the disease).

 Line 187 – “deleted” rather than “knockout” is the terminology used for adenovirus

                - DNA not cDNA

 Line 188 – “encoding” rather than “related to”

 Line 191 – delete “attack” – better to say “following direct contact with an infected animal” if that is what you mean.

               - delete “But” – not a proper way to start a sentence and not needed

               - “……. boxes from different strains of FMDV have different effects on animals immunized with adenovirus vectors”?  Is that what is meant?

 Line 194 – end sentence after “cattle”

                - “which increased the protection of species diversity”    The meaning isn’t clear.  Clarify or delete.

 Line 197 – delete “on this basis”

 Line 198 – Re-word as follows: “ ….. if additional sequences, encoding non-structural proteins, are included in the recombinant adenovirus vector genome ……”

 Line 201 – “….. which can …..” is improper sentence structure.  End the sentence with a period after “explored”.  Begin a new sentence “This FMDV antigen is expressed even more efficiently than the A24 antigen in swine.”

 Lines 202-205 – wording is not clear.  The statement here seems to contradict the statement in line 201 that the FMDV O antigens are expressed efficiently in swine. Are you now saying that fewer FMDV VLPs are produced from thew Ad5-O1Man vaccine?  Clarify.

Lines 202-205 continued – The comment about relative ability of 75S and 146S FMDV type A particles to induce neutralizing antibodies detracts the reader from the main point – delete.

                                            - Do you mean “The adenovirus vector vaccine for FMDV serotype O is inferior to the vaccine for serotype A because fewer FMDV VLPs are produced in the vaccinated animal”?

 Line 206 – delete “especially”

 Lines 207 and 208 – “vector” rather than “model”

 Line 209 – what is meant by “exogenous transcription unit”?  

 Lines 210-212 – Clarify what is meant by “enhancer promoter”

                         -  end the sentence after “promoter”. Begin a new sentence “Amino acid substitution of ….. sequence stabilizes the VLPs.”

                         - delete “But” (line 212)

 Line 214 – “to which was added a functional …..”  End sentence after “gene”.  Delete “on the basis of the Ad5 model” – it doesn’t make sense.   Reference 44 is incomplete.

 Line 216 – replace “capsid-coded region” with “sequence encoding the capsid of FMDV O Manissa”.  The reference given is 45 but the title of 44 seems relevant.

 Lines 217-220 – Do you mean “the plasmid encoding the recombinant adenovirus genome can be used to transfect E. coli and blue colonies can be selected in the presence of X-gal to identify plasmids with the genes of interest”? or is the selection done by isolating virus from blue foci of 293 cells transfected with the plasmid?   Clarify.

                         - do you mean that the plasmid itself could be used as a vaccine?

                         - “biological treatment evaluation of FMD in pigs and cattle” doesn’t make sense

 Lines 221-226 – what is meant by “diversified immunization route”?  You could simply say that recombinant human adenovirus type 5 has potential as an oral vaccine.  Consider, however, inactivation of such a vaccine in the stomach. That problem would have to be addressed.  It would be better to delete the entire paragraph since lines 222-225 don’t make sense, as written. The information in the last sentence should be moved to the beginning of this section (line 174) as something like “An adenovirus vector vaccine for FMD is licensed as a candidate vaccine in the United States for emergency use in outbreaks of FMD due to FMDV type A (references).”

 Line 227 – upper case “P” for Phage

 Line 228 – delete “also”

               - VLP singular

               - End sentence with a period after “development”.  New sentence “More of this work has been done with T7 phage.”

Line 229 – virus family names are capitalized ie Brachyviridae

Lines 232-233 – better to say “….. even when a foreign gene of more than 1 kb was inserted.”

Lines 233-235 – not clear as written.  Do you mean that the neutralizing epitope of FMDV VP1 was expressed on the surface of the phage capsid?

Line 235 – what is meant by “asymmetry of phage” here?

Line 237 – secrete not secret

               - “induce immune-related animal immune responses” doesn’t make sense, as written. Clarify.

               - “friendly relationship” isn’t scientific wording.  T7 doesn’t affect animals. It isn’t good to use the word “invade” here.  Animals and humans do have T7, at least in our intestine, but the phage doesn’t affect us directly, only by controlling levels of its bacterial host.

              - reference 48 – has the paper been published by now?  If so, update the reference.

Lines 244 – 248 – It is not clear, as written, what the nature of these DNA-loaded VLPs is. Clarify.  Being intrigued, I looked up the reference. It would be helpful to say that “the expression plasmid ….. encoding capsid proteins of FMDV, can be packaged with capsid proteins in vitro to form VLPs that can be used as vector particles to deliver the expression plasmid to target cells in the host.  Expression of the FMDV proteins then stimulates immunity through the MHC I and II pathways to produce antibodies as well as CD4+ and CD8+ T cells.”

                            - How are these vaccines administered?

Line 251 – “prolong the serological duration of the vaccine” – meaning not clear.  Do you mean “prolong the duration of the serological response” or prolong duration of the intact plasmid DNA in the vaccinated animal”?   Suggest adding “……. DNA vaccine carrier that protects the plasmid DNA from host DNAses but also enhance …..”

Line 252 – delete “In fact, the effect of ….”. Start sentence with “This vaccine is not as effective as expected and needs to be given more than three times to reach certain achieve detectable/meaningful/effective (?) antibody levels.”

Lines 256 – 260 – Two of the “problems with DNA vaccines” don’t make sense, specifically reduced immunity after IP injection across cell membrane and dilution by secretions after IN administration.  The point about protection against DNAses is good but should be made earlier in the paragraph, as suggested in the comment re line 251.

Line 262 – awkward wording.  It would be better to say “E. coli is the most common expression system for VLPs of FMDV.”

Line 265 – can expose to multiple epitopes – delete “to”

Lines 268-269 – wording doesn’t make sense, as written.  Do you mean “……… capsid, Li et al modified selected amino acids then screened the altered proteins to modified VLPs for increased hydrophobic force inside the capsid, better yield of VLPs and quality of the VLP vaccine”?  What is meant by “quality” here – immunogenicity?   

Lines 271-274 – “These live attenuated bacteria themselves are used as oral vaccines for expression of FMDV proteins in the vaccinated host.”  Is that what is meant?   Clarify.

Line 280 – “For TGE” not “In TGE”

Line 283 – delete one of the two “and”

                - baculoviruses are efficient vectors for production of viral proteins, including VLPs, in insect cells. Have any FMD vaccines been developed using baculovirus vectors?

Line 288 – replace “a new idea” with “an approach”

Lines 289-292 – not a sentence, as written

                         - “scaffolding proteins” rather than “scaffoldings”

                         - Do you mean that epitopes of FMDV have been displayed on VLPs and induce formation of FMDV-specific antibodies?   Clarify

Lines 294-295 – What is “It”? Do you mean “These VLPs”?

                         - replace “effectively absorbed” with “taken up”  (“absorbed” isn’t good terminology for uptake of particles by cells)

                        - An important goal of vaccination is to induce a memory CTL response, is it not?

                        - A reference is needed to support the statements about antibodies and cytotoxic T cells in lines 292-295.

                        - last sentence in the paragraph - Is there experimental evidence to show a CTL response to these vaccines or is it a statement about what could happen?  That point should be clarified, with appropriate references. 

Line 300 – awkward wording. Instead “….. VP1 has been used in one of the first ….”

Line 301 – “Amazingly” isn’t a scientific word – delete

               - “of cattle and swine” – delete here. Do you mean “T cell epitopes in VP4 (….) and in other structural proteins, were identified in cattle and swine”?

Line 303 – “do not involve the infectivity of the virus” – awkward wording and meaning not clear.  Delete since the point is not needed.

Lines 304-305 – awkward wording makes it difficult to understand.  Do you mean “… and the epitope sequence can be changed to represent the appropriate strain in an epidemic scenario”?

Lines 306-309 – “typical” is not an appropriate word here. Delete the first sentence of the paragraph – it doesn’t contribute information.  Start with “The first peptide vaccines represented single linear epitopes.  More recently, the multiple-antigen peptide system (MAP) (Fig. 3) has been developed.”

Lines 310-311 – Clarify wording as follows:  MAPs (plural) are dendritic polymers with lysine dendrimers exposed as at the core and dendritic arms to with multiple or more antigenic epitopes

Lines 312-314 – For clarity, re-arrange wording as follows:  “…… has better is more immunogenic and moderately than a simple linear juxtaposition of B and T epitopes and is moderately resistant to enzyme digestion.  In contrast, VP1 is degraded easily by trypsin.”

Line 316 – delete “different”.  The word here is redundant since it is included by the term “antigenic diversity”.

Lines 322-333 – For clarity, re-arrange wording as follows ”…….. were increased to give stronger protection when swine were infected (not attacked).”

Line 323 – “with the exception of” doesn’t make sense here.  For clarity, it would be better to say “Along with adenovirus vector vaccines, this peptide vaccine is one of the few to show that a single dose can replace conventional FMDV vaccines for protection against disease”.

Lines 325-326 – “single-dose vaccines”  ie insert “vaccines”

                         - help not helps

                         - “facilitates” rather than “enables”

Line 327 – re-word as follows: “Emergency vaccines need to provide a fast, low-cost and flexible response to the virus.”

Line 329 – “key” rather than “decisive”

Line 330 – does (56-70) mean (aa 56-70)?  Clarify.

Line 331 – re-word as follows:  “……. which were inoculated with …. produced the same level of neutralizing antibodies and IFN-gamma as B2T-3A.”   “high frequency of IFN-gamma” doesn’t make sense.     

Line 333 - “even B2T-3D preferentially recognized the T3D peptide” doesn’t make sense since B2T-3D (the vaccine) can’t recognize a peptide.  Do you mean that “inoculation with B2T-3D induced even more IFN-gamma”?

Line 334 – re-word as follows:  “ ….. induced by B2T-3D was higher than that included by B2T-3A.”

Line 335 – For clarity, re-word as follows:  “The B2T-TB2 peptide, with six epitopes, is formed by cross-linking two B2T molecules and can induce ….”

Lines 337-338 – Humoral immunity to CSFV is interesting given that CSFV is in a different virus family than FMDV.  The classification of CSFV should be included here.

Lines 341-356 – The final sentence of this paragraph, modified as follows: “Recent findings show that specific MHC alleles and haplotypes ….” would be helpful to the reader if moved to the beginning of the paragraph. It follows logically from the previous paragraph. The authors could then continue with line 346 onwards, starting with “Porcine MHC …...”.

                         - delete lines 341 – 346 (after “polymorphic”).  The content is confusing, as written, and not needed.

Lines 358-360 – Delete the first two sentences of this paragraph.  The wording, as written, is awkward and detracts from the remainder of the paragraph.  Start paragraph with “The development of …”

Lines 360-363 – awkward wording. Re-word as follows: “Successful vaccines depend on safe and effective adjuvants as well as appropriate delivery systems and route of administration.

Line 364 – It’s not clear what is meant by “more mature adjuvants”.  Do you mean “Adjuvants used to date include …..”?  What are the adjuvants that are commonly used?   

                - delete “etc.”

Lines 370 and 372 – for “Oil-adjuvant”, do you mean “Oil adjuvants”?

                                 - delete “actually”

                                 - do you mean that “Oil adjuvants are more effective than other adjuvants in swine, causing stronger immune responses”?

Line 371 – “they” rather than “it”

Line 372 – “induced strong immunity” rather than “had strong immunity”

Line 373 – delete “even”

Lines 375-376 – delete the final sentence of the paragraph – it doesn’t contribute new information and detracts from the earlier part of the paragraph.

Lines 377-386 - Having looked at reference 28, I see that those authors described the addition of sucrose to culture medium during amplification of virus for increased stability of the virus during repeated freeze-thaw cycles d during storage.  Unless sucrose is added to the vaccine itself, it wouldn’t protect the virus once the virus was mixed with the oil adjuvant.  These sentences about sucrose in culture fluid are not relevant to stability of virus in oil adjuvant mixture and should be deleted.

                         – move “The degradation of antigen is associated with protein denaturation at the oil-water interface” to the end of the previous paragraph, followed by “The new water-in-oil emulsion ……. increased [79].”

Line 388 – antibody-mediated

Line 390 – awkward wording.  Re-word as follows: ”Saponin was combined with aluminum hydroxide (alum) to effectively ….. immunity [80,81].”

Lines 392-396 – awkward wording.  Re-word as follows:  “Saponins also provide an advantage when mixed with oil adjuvant. In cattle, a saponin-oil emulsion increased the level of neutralizing antibody with a lower dose of antigen (146S particles) [82].  Reducing the required dose of viral antigen represents a significant cost saving in vaccine production.”

                         - it would be helpful to include details such as specific dose of antigen and neutralizing antibody levels that were achieved from reference 82 so that the reader doesn’t have to look up the paper to get that information.

Line 397 – It is more professional to say “Humoral immunity to FMDV develops rapidly but is short-lived ….”

               - what is meant by “iFMDV”?

Line 399 – what is meant by “model antigens”?

Line 402 – awkward wording – instead, “the direction of adjuvant development”

Line 403 – “to antigens” rather than “of antigens”

Line 404 – “ingested” means “by mouth”, as with oral vaccines.  Is that what is meant here?

                - “in association with MHC molecules” rather than “to express MHC molecules”

                - a general immunology reference is needed to support the statements in this paragraph

Line 406 – end the sentence after “process”. Continue with a new sentence as follows: “Mature DCs are recognized by CD4+ and CD8+ T cells, resulting in both humoral and cellular immunity.”

Line 414 – “a” rather than “A”

               - for clarity, say “This modification” rather than “It”

               - HBHA of Mycobacterium tuberculosis (it is meaningful to include the source of the HBHA)

Lines 417-18  – awkward wording.  Re-word as follows:  “ Chinese herbal medicine, with a long history and few side effects, cannot be ignored when considering adjuvants.

Lines 419-422 – awkward wording.  End the sentence with a period after “enhancers”.  Begin a new sentence as follows: “The carbohydrate groups on the saponin molecules could react with receptors on the antigen-presenting cells (APCs) to promote a Th1 response [88].

                         -  what is meant “by foreign antibodies”?

Lines 417-431 – for each study mentioned, what was the form of the antigen?  Inactivated virions?

Line 423 – write out the full name for C. deserticola.  The name should be italicized.  

               - insert a space before “(CPCD)”

Line 424 – for clarity, change “It” to “CPCD”

Line 426 – italicize “Artemisia rupestris

Line 428 – “immune persistence” is a strange term. Do you mean “longterm immunity”?

Line 429 – italicize “Achyranthes bidentata

Line 431 – “preparation” rather than “system”

Line 432 – Re-word as follows: “Non-coding RNA (ncRNA) of FMDV also has potential as a vaccine adjuvant”. 

Line 433 – Re-word as follows “ ….. short RNA sequences whose structures are similar to those within the non-coding regions of the FMDV genome.”  Delete the remainder of the sentence – there is no need to define FMDV at this point in the paper.

Lines 435-438 – the meaning is not clear.  After looking up the paper, I suggest “The 3’ NCR, as well as the IRES and S domains from the 5’ NCR, induced stronger and longer-lasting B and T cell responses, enhancing protection by inactivated FMD vaccine in swine [92].”

Line 440 – “to” rather than “of”

                - It would be good to remind the reader at this point what MAP stands for ie “MAP (multiple antigen peptide) vaccine”

Line 452 – insert “nanoparticles” ie “ ….. silica nanoparticles (DMSNs) …..”

                - “have” rather than “has”

                - “greater” rather than “stronger” loading capacity

                - should the reference be “An et al” since “Weitang” appears to be the first name?

Lines 453-456 – For clarity, re-word as follows: “An et al used DMSNs with B2T synthetic peptide and showed prolonged release when the complex was coated with serum proteins.”   According to reference 94, the 61% and 80% refer to the proportion of loaded peptide that was released, not to the amount by which peptide release was increased from the coated DMSNs. The precise amounts of peptide released from the DMSNs aren’t needed here.   

Line 458 – “ingeniously” is used twice; delete one of them

Line 465 – add “it” to become “….. because it can be …….”

Line 467 – what is iFMDV (also used back in line 397)?

Line 472 – “to” rather than “of”

Line 473 – “due to …….. high biocompatibility [99]”     This part of the sentence doesn’t make sense, as written – is something missing?

Lines 474-477 – These lines don’t make sense, as written.  They need to be re-written for clarity.   

                         - what is the reference for Zheng et al?  It seems to be missing from the reference list

Lines 478-480 – a reference is needed to support this statement

Line 481 – “and” rather than “which”

Line 483 - add “release” ie “….. and the release of immune-related cytokines [100].”

Lines 485-487 – “….. LDH has a layered structure”.  Delete since it’s obvious.

                         - what is meant by “lamination” here?   Can you say that “…….. (LDH) has an overall positive charge”? 

          - The rest of the sentence doesn’t make sense, as written.  Reference 101 wasn’t sufficient to explain the material since it didn’t provide the details mentioned here.  An additional reference is needed.    

Line 493 – delete “been”

Line 496 – systems (plural)

Table 2 – re sucrose – viral capsid protein and protein.  What is the second protein here?   

              - re saponin – Th1/Th2 “response” rather than Th1/Th2 “immunity”

                                    - “permeabilizes cell membranes” not “is permeable to cell membranes”

              - re cationic liposomes…..  – shouldn’t the vaccine be VLPs not inactivated viruses since VLP is specified under the “Mechanism” column?

              - re HBHA – insert “HBHA” for clarity ie “ …. was combined with HBHA, a novel TLR4 agonist”

              - re CVC1302 – “induce bone marrow” doesn’t make sense.  Re-write for clarity.

                                      - “increase” rather than “transform”

               - re Chinese herbal medicine – the corresponding vaccines are listed as “inactivated vaccines”. That term doesn’t make sense.  Do you mean “inactivated viruses”?   The same question applies to listed vaccines corresponding to chitosan and to gold nanocages as adjuvant/delivery systems.

- re ginseng – what is meant by “foreign antibodies” (also in line 421)?

              - re CPCD – C. deserticola should be italicized. Write out the full name, to be consistent with full names for other organisms

                               - “DCs were activated” not “DCs was activated”

              - italics for Artemisia rupestris and Achyranthes bidentata

              - re Rnas – use “RNAs” rather than “Rnas”

                              - in the Mechanism column, clarify as follows: ”IRES, S and 3’NCR domains transcribed in vitro from plasmids”

                             - reference 92 used inactivated viruses not VLPs

              - re mesoporous silica – are these carriers not loaded with peptides (B2T) rather than inactivated viruses?

              - re chitosan – clarify as “…. configuration and deformation of the vaccine particles can increase the contact area with cells”

              - re gold nanocages – isn’t reference 100 about VLPs, not inactivated virions?  

Line 506 – “which have” rather than “which has”

Line 507 – delete “the” ie “replication” rather than “the replication”

               - what is meant by “host selection process”?   The meaning is not clear but the term is not necessary.  Could you say “……. due to mutation and recombination during the process of replication”?

Line 508 – “mutation rate of protein” is not correct.  Instead, say “VP1 is most frequently affected by mutation whereas VP4 is more conserved.”

Conclusion – condense and re-word for clarity and impact.

                    - A major challenge with FMD vaccine appears to be that there are multiple types of FMDV that don’t cross-react immunologically. Emphasize that point to the reader without repeating information already presented in the Introduction. A single sentence is sufficient.

                   - Given that infected swine exhale large amounts of virus that can travel up to ????? miles or km in the air, it is important that vaccines be effective in swine. 

     - Whereas the established vaccines, in use since ?????, are inactivated virus particles, the newer vaccines in use represent subviral components in the form of adenovirus vectored vaccines, VLPs and synthetic peptides. In all cases, the immunogen is intended to elicit a protective response among multiple strains of FMDV.

                   - As the authors point out, duration of immunity is also a challenge.  Existing inactivated virus vaccines require repeated dosing twice a year.

                   - With newer vaccines that protect animals against multiple virus strains, hopefully for a longer period of time, we are better prepared for the next outbreak of FMD. Continued research should lead to further improvements in protection against FMD.  

Author Response

Thank you very much for your advice, between the lines can feel you to modify my article has carried on the very detailed, I also tried to digest you give each a piece of advice, but there might be some omissions place or change is not good place, also please the teacher more guidance, the growth of these proposals to my writing skills are very effective. Please find my itemized responses in below and my corrections in the re-submitted files.

Line25, Sorry, I did not find "{" in the original manuscript I downloaded, is it the display problem?

Line117, The complete virions of 146S are referred to here.

Line366,As I explained earlier: iFMDV is inactivated FMDV(To view:Line169)

Line471-431, The antigens corresponding to each study, I have listed in the table below which vaccine the study was focused on.

We would like also to thank you for allowing us to resubmit a revised copy of the manuscript. We hope that the revised manuscript is accepted for publication in the Journal of VACCINES.

Sincerely,

LU Zhimin

Reviewer 2 Report

In this manuscript “Development of foot-and-mouth disease vaccines in recent years”, the authors introduced the recent development of vaccines for foot-and-mouth disease, including development of different vaccine formats, different antigens, different adjuvants, etc. Overall, this is an interesting and timely review. However, this manuscript is not well written. Some description is too vague, lacks details. Grammar issues exist throughout this manuscript. Therefore, a major revision is needed to improve the quality of this manuscript.

Major issues:

1. Line 30-31, “Although cattle are the main host, some strains are also sensitive to swine”, grammar issue. Change it to “although cattle are the main host, swine are susceptible to some strains too”?

2. Line 33-38, the authors mentioned the main host is cattle, then why did the authors start to introduce the pathogenesis in swine, instead of the pathogenesis in cattle?

3. Line 48, please clearly state the length of the genome, this is a critical piece of information about the virus.

4. Line 90, “it shortened the time of the vaccine”, I don’t understand this sentence. Does it mean the production time of vaccines or the time needed to fully vaccine a host? Please rephrase.

5. Line 99, please explicitly describe the definition of these three terms, the "white zone", the "gray zone" and the "black zone".

6. Line 114, “which is not conducive to the full development of T cells”. I don’t understand this sentence, please rephrase.

7. Line 115-116, please check the grammar.

8. Line 125, “it can be said that this serotype is now extinct outside the laboratory”, please explicitly state which authority or organization made this claim.

9. Line 127, please explain in detail what r1 value is in the manuscript.

10. Line 153-157, this paragraph is too vague and lacks details.

11. Line 162, please clearly define what virus-like particles are.

12. Line 235-236, “The asymmetry of phage can enhance the immune response of T helper cells”, I don’t understand the meaning of this sentence. Please explain in detail why asymmetry of the phage would enhance T cell responses. And references are needed.

13. Line 322-325, please check the grammar.

14. Line 330, “T cell epitopes of FMDV non-structural protein 3D (56-70) in FMDV”, T cell epitopes are typically several amino acids longer than that. Could the authors use one or two sentences to briefly describe in this manuscript why the 3D epitope is only 5-aa long?

15. Line 342-346, there are several severe grammar issues in this paragraph. In addition, MHC associated epitopes are T cell epitopes, the description here is not scientifically sound. Please rephrase this paragraph.

16. Line 388, “the typical antibot-mediated Th2 response rather than cell-mediated Th1 immunity”, I don’t understand the meaning of this.

17. Line 405-406, “Some agonists have been shown to induce DCs maturation during this process, which is recognized by CD4+ and CD8+T cells”. Please check the grammar.

18. Line 418, “Chinese herbal medicine is natural and has few side effects with a long history”, references are needed here to support the claim. None of the references in this paragraph supported that usage of Chinese herbal medicine as adjuvant had few side effects in authority-approved vaccines with “a long history”. All studies listed here are pretty new pre-clinical studies.

19. Line 495, “improvement of CD4+ and CD8+ responses in body fluids and cells”. Please check the grammar.

20. The conclusion section is too tedious and lacks focus. Please shorten this part and make it more concise.

Author Response

Thanks very much for taking your time to review this manuscript. I really appreciate all your comments and suggestions! Please find my itemized responses in below and my corrections in the re-submitted files.

Line 33-38,the pathogenesis of the pig was introduced because this article wanted to highlight that FMD affects pigs strongly, although cattle are the main host of FMD. In many other aspects, this article also puts forward prevention and control measures against foot-and-mouth disease from pigs.

Line 230-231,References 49, 50 are newly cited references in which it is mentioned that the use of bacteriophage asymmetry can better stimulate the immune response of T cells.

Line 319-320,The authors synthesized 92 synthetic peptides covering 3D amino acid sequences in the article “Immunogenicity and T Cell Recognition in Swine of foot-and-mouth Disease Virus Polymerase 3D”. Moreover, the author only did the corresponding experiments for 3D (56-70) in the literature "Immunogenicity of a Dendrimer B2T Peptide Harboring a T-cell Epitope From FMDV non-structural Protein In 3D ". It may be that the peptides synthesized from this part of the 3D epitope show a higher level of induction.

We would like also to thank you for allowing us to resubmit a revised copy of the manuscript. We hope that the revised manuscript is accepted for publication in the Journal of VACCINES.

Sincerely,

LU Zhimin

Round 2

Reviewer 1 Report

General comments for the authors

This revised version shows considerable improvement over the first version but work is still needed to make it publishable.

In several places, the reference doesn’t match the statement being made.  Those situations are addressed in the specific comments to follow.

The following references are incomplete ie no date or journal information: 34, 35, 36, 38, 41, 42, 44, 67.     

Specific comments to be addressed by the authors

Line 49 – change to “…… (ORF) encoding a polyprotein that is processed into mature polypeptides.”

Line 51 – delete “which”

Line 71 – delete “Among them”

Line 82 – change to “provides new approaches for development of FMD vaccines”

Line 87 – Inactivated virus vaccine

Lines 93-94 – change to “ ….. emergency vaccination in the event of an outbreak was approved [16].”

Lines 95-96 – change to “FMDV is highly sensitive to neutralizing antibodies produced in response to whole inactivated viruses and antibody titres are closely related to protection.”

Lines 99-101 – change “area” to “zone”

Lines 103-106 – delete “Swine also …… identified”

Line 114 – delete “also”

                -  delete “the most effective”

Line 115 – replace “animals” with “guinea pigs”

                - replace “but the effect was weaker” with “but less effectively than complete virions (146S) [20].”

Lines 120-121 – delete “They should be”

Lines 129-130 – replace “VNT” with “virus neutralization (VN) titre”

                       -  replace “The r1 value ………. homologous sera” with “The r1 value (relationship coefficient) is the ratio of serum VN titre against a heterologous strain to serum VN titre against the homologous strain.”

Line 134 – delete “to infect susceptible swine and”

Lines 145 – 146 – do you mean “They identified the neutralizing antibody in a complex with the virus, by cryo-electron microscopy at high resolution.”?

Line 149 – change “Currently ………. BHK-21 cell”  to “Currently, the continuous cell line for production of FMDV is BHK-21”

Lines 154-155 – delete “or media to modify it”

                          - replace “another one is through …..” with “Another approach is through cellular modification …..”

Line 156 – change “by integrins” to “by expressing integrins”

Line 158 – replace “production” with “yields”

Line 161 – delete “in inactivated virus”

Line 164 – delete “autonomously”

Line 166 – replace the first “vaccine” in this line with “virus”

Line 171 – VLP not VLPs

Line 178 – replace “inactivated vaccines” with “inactivated virus vaccines”

Line 180 – delete “of all the”

Line 181 – replace “The second is that” with “Second,”

                - immune responses (plural)

Line 183 – replace “for recognition by CD8+ cytotoxic cells” with “and stimulate a CD8+ cytotoxic T cell response”

Line 187 – delete the second “vaccine”

Line 188 – replace “whose DNA encodes sequences” with “with sequences”

Line 189 – replace “promotes” with “mediates”

Line 196-197 – replace “Many researchers …… Ad5-A24 [39]” with “Pena et al modified the Ad5-A24 vector by including FMDV sequences encoding non-structural proteins [39].”

Lines 202-203 – delete “some problems were discovered:”

Lines 210-212 – replace “whose objective is to stabilize the expression of ….. VLPs” with “stabilizes the VLPs”.

Lines 213-218 – delete

Lines 219-221 – delete

Line 223 – change “VLPs” to “VLP”

Line 224 – delete “Phage”

Lines 227-235 – replace “At the same time ………… vaccine preparation” with “Peptides of FMDV VP1, when displayed on the phage capsid, induced a specific antibody response in mice [47, 48], suggesting that such particles may have potential as a vaccine. The stability of T7 phage would be a distinct advantage in reducing the storage and shipping costs of such vaccines.”

         By way of comment, not for incorporation into the paper:   References 49 and 50 are not relevant.  Neither reference mentions the asymmetry of phage nor does either reference support the statement. Reference #49 is about filamentous phage displaying a peptide from yeast not about T7 phage displaying a peptide of FMDV.  Reference 50 is not about vaccine at all.  After reading ref 48, I see that the sentence about asymmetry of phage is copied directly from ref 48 but it doesn’t make sense there either.  The references given in #48 to support that statement are the ones you list as #49 and #50.  They are not relevant to FMD vaccine.

Lines 238-240 – Replace “The virus-like particles formed by ….. empty capsid [51],” with “The pcDNA3.1/P12A3C expression plasmid encoding proteins of FMDV”

Line 240 – delete “which”

Line 243 – immediately after “……….. CD8+ T cells.”, insert the following “These DNA-loaded VLPs function as a DNA vaccine carrier that protects the plasmid DNA from degradation by host enzymes. Moreover, they enhanced the level of specific antibody and prolonged its duration compared to VLPs without DNA and to plasmid DNA alone in a guinea pig model [51].  Plasmid DNA vaccines given alone are not highly effective, requiring three doses to achieve effective antibody levels in pigs [53,54], though co-injection with a plasmid encoding GM-CSF induced a stronger response in cattle [52] and swine [53].”

Line 243-249 – delete “Tandem expression …… antibody levels [52-54].”

Lines 249-252 – meaning is not clear and there is no reference – delete.

Line 254 – “Escherichia coli” should be italicized

Line 260 – change “capsid” to “VLPs”

Line 261 – delete “altered proteins to”

Lines 263-264 – “Salmonella typhimurium”, “Lactococcus lactis” and “Bacillus subtilis” must all be italicized.

                          - a reference is needed for Bacillus subtilis

Line 275 – change “[59]” to “[59, 60]”

Lines 279-283 – replace “Using hepatitis B ……… antibodies” with “Epitopes of FMDV displayed on VLPs composed of hepatitis B core protein or HIV Gag polypeptide or rabbit hemorrhagic disease virus (RHDV) VP60 induce formation of FMDV-specific antibodies [61, 62, 63, 64].”

Line 283 – replace “These” with “Similar”

Line 285 – continue with “It may be expected that VLPs with T-cell epitopes of FMDV could induce a cytotoxic T-cell response as seen for other T-cell epitopes expressed as part of RHDV VLPs [65, 67].”

Line 291 – delete “were identified”

Line 297 – delete “to”

Line 298 – delete “has better”

Line 299 – delete “and moderately”

Line 303 – FMDV not FDMV

Lines 311-313 – delete “This vaccine is in addition to ….. vaccines”.

Line 316 – replace “facilitates” with “facilitate”

Line 317 – change “in emergency situations. After all, …… the virus” to “in emergency situations which require a fast, low-cost and flexible response to the virus.”

Line 320 – delete the first “FMDV”  

                - change “(56-70)” to “(amino acids 56-70)”

Line 324 – change “included by” to “induced by”

Line 327 – replace “showed” with “induced”

Lines 327-328 – replace “the pestivirus of Flaviviridae” with “a pestivirus within the family Flaviviridae”

Line 334 – replace “Based on this, they” with “These alleles”

Lines 335-338 – references are needed to support these statements

Line 341 – delete “(eg oral administration)”

Line 342 – insert comma between aluminum hydroxide and saponins ie “aluminum hydroxide, saponins”

Line 343 – replace “Quila” with “Quil-A”

Lines 352-355 – replace “the degradation ……. 146S particles [82].” with “degradation of immunologically active 146S particles at the oil water interface [82].”

                         - delete “The degradation of antigen ……. virus [83].”   Reference 83 is not relevant. 

Lines 366-367 – replace “Although …… immunity [90]” with “Although the immunopotentiator CVC1302 can enhance humoral immunity [90]”   

Lines 369-370 – replace “In some cases ……… shifted” with “Since an enhanced cellular immune response can be more beneficial in some cases, the direction of adjuvant development has been shifted”

Line 372 – replace “ingested” with “taken up”

Lines 387-389 – replace “The carbohydrate groups ……. Th1 response [95].” with “Carbohydrate groups on the saponin molecule can interact with receptors on antigen-presenting cells (APCs) to promote a Th1 response [95].”

Lines 389-390 – replace C. deserticola with “Cistanche deserticola  (italicized)

Line 393 – optimal dose of 400 micrograms – in what animal?

Line 415 – references are needed to support the statements in this paragraph

Line 421 – delete “ingeniously”

Line 424 – delete “a certain amount of”

Line 440 – replace “swine” with “pig”

Lines 436-441 – replace “Zheng et al …… mucosal sites” with “Zheng et al used chitosan-coated PLGA-NPs mixed with amino-functionalized mesoporous silica nanoparticles, loaded with MDV recombinant plasmid, for intranasal vaccination along with CpG oligodeoxynucleotides encapsulated in chitosan-coated PLGA-NPs as adjuvant [107]. Systemic and mucosal immune responses, including specific IgA, were induced in guinea pigs.  Challenge experiments showed protection against systemic spread of virus in vivo following injection of virus into one leg [107].”

Lines 441-442 – delete “In addition ……….. advantages”.

Lines 447-452 – By way of comment:  This paragraph was extremely difficult to understand.  This reviewer knew nothing of LDHs and had to find a reference to give me sufficient background to make sense of what was being said.  Mishra G et al (2018) was very helpful.  Much of what the authors say in the manuscript being reviewed seems to be copied from Wu et al (ref 110).  That description isn’t particularly helpful for the reader; the information about results (lines 449-452) refers to other studies mentioned by Wu et al but these results are not from the experiments done by Wu et al and reported in ref 110.  I have re-written the paragraph, with the intention of capturing what the authors were trying to say – see below:

          - replace with “Layered double hydroxides (LDHs) are structures with alternating layers of hydroxide molecules separated by interspace layers filled with anions and water.  These anions can be exchanged for other molecules, including biological molecules, meaning that LDHs can be loaded with biological molecules and used as delivery vehicles [Mishra G et al, Appl Clay Sci 153, 172-186 (2018)].  The properties of LDHs, including sustained delivery of their load, make them attractive as potential vaccine carriers [Chen W et al (2018), Small 14, 1704465; Yan SY et al (2018), Front. Pharmacol 20].  When used to immunize mice by subcutaneous injection, LDHs loaded with inactivated FMDV particles induced high and sustained antibody levels, following an initial delay [Wu P et al, 2020]. In pigs, the LDHs loaded with inactivated FMDV particles induced specific neutralizing antibody at levels comparable to levels achieved with Montanide ISA-206 adjuvant [Wu et al, 2020].”

Table 2 – for Chinese herbal medicine and for nanoparticle polymers - the corresponding vaccine should be “inactivated virus” rather than “inactivated viruses vaccines”

              - for ginseng - do you mean “antigens” rather than “antibodies”?

              - for CPCD – Cistanche deserticola (italicized)

              - for RNAs – delete “produced from short RNA transcripts”

                                 - ref 99 – the abstract talks about inactivated viruses not VLPs.  Change “VLPs vaccine” to “inactivated virus”

              - for mesoporous silica – only ref 103 used inactivated virus.  101 used peptide B2T, 102 used an anti-tumour drug (not relevant here), 107 used plasmid DNA.  List the appropriate kind of vaccine in the final column.

              - for chitosan – refs 104 and 105 are about inactivated virus.  Ref 106 is about DNA vaccine delivered with PLGA nanoparticles with cytokines, not chitosan, as adjuvants. Delete ref 106 here.  The last column should list inactivated virus only.

              - for LDH – ref 110 is about using LDHs with inactivated virus, not with VLPs. Change the fourth column to “inactivated virus”

Line 470 – replace “appears to be” with “is”

Lines 471-472 – delete “Emphasize that point ……….. in the Introduction”. That was a comment to the authors, not meant to be incorporated into the text.

Line 473 – delete “and have a magnifying effect on FMDV”

Line 474 – delete “Therefore ……….. targeted vaccine.”

Lines 479-481 – delete “By mimicking ……. real virus hits”

Lines 481-484 – replace “as a kind of an effective …. to maintain protection” with “Whereas existing inactivated virus vaccines require repeated dosing, it is desirable that vaccines would provide protection after a single vaccination.”

Lines 484-490 – delete “It is also important …….. simultaneously”

Lines 491-494 – delete “In a word ………. more innovative vaccines”

Line 494 – delete “hopefully”

Line 547 - immunogenicity

Author Response

Dear reviewer, I have revised each of your suggestions. Thank you for your review!

Reviewer 2 Report

The authors indeed spent efforts trying to improve the manuscript. However, some basic immunology concepts or description here are still scientifically misleading. Besides, I didn't see a point-by-point response to my review opinion, after I spent a decent amount of time reviewing the manuscript and writing the comments.

Major issues:

Issue No. 14, from the authors' response it seems that peptide 3D, which is 5-aa long, is part of an epitope, but not an epitope recognized by T cells. T cell epitopes are longer than 5-aa due to MHC restriction. Please check basic immunology concepts and make corrections in the manuscript accordingly.

Issue No. 16, the authors made no change. Th2 response is not necessarily mediated by antibody, cell mediated immunity or cellular immunity refers to general T cell responses including both Th1 and Th2. Please check basic immunology concepts and rephrase this sentence.

Issue No. 20, in the revised conclusion, line 469-471, "A major challenge with FMD vaccine appears to be that there are multiple types of FMDV that don’t cross-react immunologically", this is not scientifically sound, probably due to grammar issue. Change it to "A major challenge with FMD vaccine appears to be that the available vaccine targeting one serotype of FMDV doesn't generate immune memory cross-reacting against other serotypes of FMDV"?

Author Response

Dear reviewer, I have revised each of your suggestions. Thank you for your review!

Issue No.14,I have made the corresponding changes.

Issue No.16, I can't remember where I first read the relevant report. Irrelevant statements were therefore removed, leaving only the facts mentioned in the cited literature.

Issue No.20,I have made the corresponding changes.

Round 3

Reviewer 1 Report

The manuscript is very much improved over the previous version. A few minor corrections remain.  These are listed below:

Line 48 – change “including” to “includes”

Line 99 – change “white area” to “white zone”

Line 102 – change “gray area” to “gray zone”

Line 129 – delete the first “strain” in line 129

Line 131 – “to mutate” instead of “mutate”

Line 184 – “with sequences” instead of “sequences”

Line 193 – “encoding additional non-structural proteins” instead of “encoding non-structural proteins”

Lines 274-275 – delete “or more”

Lines 297 and 298 – “induced by” instead of “included by”

Line 310 – delete “The analysis of “

Line 364 – replace “at the optimal dose of 400 ug in cell experiments” with “in mice”

Line 408 – FMDV not MDV

Table 2 – re first three entries for Chinese herbal medicine – in each case, the last column should say “inactivated viruses” not “inactivated vaccines”

              - re non-coding synthetic RNAs – the last column should say “inactivated viruses”

              - re mesoporous silica – the last column should say “inactivated viruses”.  Delete “vaccines”.

              - re chitosan – replace “VLP vaccine and inactivated vaccine” with “VLP vaccine and inactivated viruses”

              - re layered dihydroxide – last column should be “inactivated viruses”

                                                     - for mechanism of action, replace the existing words with “These particles, with interspace layers that can be loaded with antigens, provide improved and sustained delivery of antigen in vivo [107].”

Line 451 – “against” instead of “among”

Line 508 – “immunogenicity” instead of the word as printed

Author Response

Thank you very much for your valuable suggestions. I have made corresponding modifications to each of your suggestions. Please criticize and advise me.

Sincerely,

LU Zhimin

Reviewer 2 Report

All my previous concerns resolved.

Author Response

Thank you very much for your advice.

Sincerely,

LU Zhimin
